# Learning Debiased and Disentangled Representations for Semantic Segmentation

**Sanghyeok Chu**     **Dongwan Kim**     **Bohyung Han**
ECE & ASRI, Seoul National University
{sanghyeok.chu,dongwan123,bhhan}@snu.ac.kr

## Abstract

Deep neural networks are susceptible to learn biased models with entangled feature representations, which may lead to subpar performances on various downstream tasks. This is particularly true for under-represented classes, where a lack of diversity in the data exacerbates the tendency. This limitation has been addressed mostly in classification tasks, but there is little study on additional challenges that may appear in more complex dense prediction problems including semantic segmentation. To this end, we propose a model-agnostic and stochastic training scheme for semantic segmentation, which facilitates the learning of debiased and disentangled representations. For each class, we first extract class-specific information from the highly entangled feature map. Then, information related to a randomly sampled class is suppressed by a feature selection process in the feature space. By randomly eliminating certain class information in each training iteration, we effectively reduce feature dependencies among classes, and the model is able to learn more debiased and disentangled feature representations. Models trained with our approach demonstrate strong results on multiple semantic segmentation benchmarks, with especially notable performance gains on under-represented classes.

## 1   Introduction

Semantic segmentation is a pixelwise classification task that is applicable to a wide range of practical problems including autonomous driving, medical image diagnosis, scene understanding, and many others. Ever since deep neural networks have been adopted for semantic segmentation [1–5], its accuracy has increased gradually with the introduction of stronger network architectures, improved training schemes, and large-scale datasets.

Despite such phenomenal achievement, semantic segmentation approaches still suffer from the chronic limitations caused by class imbalance and stereotyped scene context in datasets. The class imbalance issues are prevalent in semantic segmentation, where certain classes occupy larger area and/or appear more frequently than others. For example, the "road" class in the Cityscapes dataset [6] accounts for over 36% of pixels of all training images while the "motorcycle" class does only 0.1% of pixels. Furthermore, the scene context is not sufficiently diversified in training datasets. For instance, the objects in the "motorcycle" class is always observed in the vicinity of the "road" class, and the spatial layout of the two classes are almost identical in all images. Consequently, the models often learn substantially correlated representations and fail to accurately delineate the boundaries in the absence of frequently co-located objects.

These issues cause the model to learn suboptimal feature representations, especially for classes that are under-represented. These feature representations are often entangled with one another, causing confusion between classes, and are also susceptible to encode adverse biases as well, which may be detrimental to the model's overall performance. On the other hand, for classes with abundant labels, the model is able to leverage the large variations in the data and learn more robust feature

35th Conference on Neural Information Processing Systems (NeurIPS 2021).

representations that are less dependent on the presence of other classes. As such, the model exhibits strong performance on classes with sufficient labels but weaker performances on rare classes.

We propose DropClass, a simple yet effective training scheme for semantic segmentation that aims to alleviate the vulnerability of dataset bias and feature entanglement. Given an input example, we first extract a class-specific feature map for each class using Grad-CAM [7]. Then, we drop the feature map corresponding to a randomly sampled class and aggregate the remaining feature maps to generate the predictions. Finally, we employ our proposed loss functions to facilitate training. This procedure is somewhat similar to Dropout [8], but is specifically designed with a different goal in mind; while Dropout aims to reduce co-adaptations among features, DropClass aims to disentangle the class representations and reduce the underlying biases that arise from inter-class relationships. Our training scheme requires no additional data or annotations. More importantly, since DropClass is model agnostic, it can be plugged into any existing network architecture straightforwardly.

Our contributions can be summarized as follows:

- Our work takes the first step to tackling dataset biases in the semantic segmentation task. We describe how dataset biases in semantic segmentation are different from those in classification, and devise a benchmark to better analyze the biases found in segmentation models.

- We propose DropClass, a model-agnostic training scheme that results in learning more debiased and disentangled feature representations for semantic segmentation.

- Our experimental results show that the model trained with DropClass exhibits stronger performance compared to the baseline across multiple network architectures and datasets. Furthermore, we observe significant performance gains especially on under-represented classes, and provide analysis that further validate the effectiveness of our training scheme.

## 2 Proposed Method

We first define the semantic segmentation task and its relevant notations. For a dataset with a set of classes, $\mathcal{C}$, we sample an input image $x \in \mathbb{R}^{h \times w \times 3}$ and its corresponding label $y \in \mathbb{R}^{h \times w \times |\mathcal{C}|}$. The feature extractor $g(\cdot)$ takes $x$ as input and produces an intermediate feature representation $A \in \mathbb{R}^{h_1 \times w_1 \times k}$, where $k$ denotes an arbitrary channel size of the feature map that depends on the choice of network architecture. The feature map $A$ is then fed into the classifier, $h(\cdot)$, to produce $\hat{y} \in \mathbb{R}^{h \times w \times |\mathcal{C}|}$, which serves as the final output of the model before the softmax operation, *i.e.,* the logits. Altogether, we denote the entire model as a composition of the feature extractor and classifier: $\hat{y} = h(A) = h(g(x))$. Note that throughout the paper, we use subscripts to refer to specific spatial and channel locations on tensors.

### 2.1 Motivation

Dataset bias issues are introduced in a plethora of literature dealing with classification problems [9–13], where the primary concerns are class imbalance in the dataset and spurious correlation between attributes in images. Although the dense prediction tasks including semantic segmentation may suffer from additional challenges related to dataset bias, their potential limitations have still been hardly discussed. Hence, this work first points out a critical drawback exposed to semantic segmentation algorithms caused by the stereotyped co-occurrence of multiple classes as well as the issues inherent from the classification tasks.

We argue that, besides the class imbalance and attribute correlation issues, inter-class relationships with spatial co-occurrence or feature similarity could be the culprit of dataset biases in semantic segmentation models. Consider the class relationship between the "car" and "road" classes in road scenes segmentation. The two classes have a strong spatial correlation since the vast majority of "car" samples will be spatially located above the "road" class. Models can easily exploit this spatial correlation in the process of maximizing the pixel-level accuracy. However, this could have adverse effects when the model encounters an image where "cars" do not appear above the "road" class. Moreover, the representations of a class are often entangled with one another since the features of co-occurring classes in an image are mixed in the convolution layers, and this tendency is aggravated with the semantic similarity between classes, e.g., "bicycle ↔ motorcycle" and "person ↔ rider". In both cases, under-represented, rare classes typically become the victims of the feature entanglement.

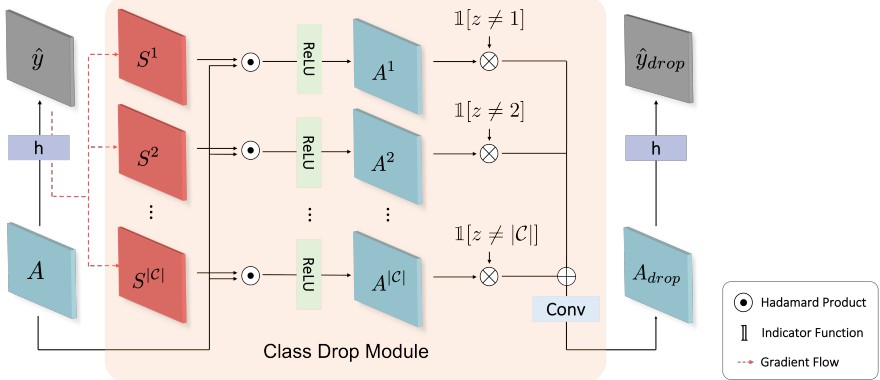

Figure 1: An illustration of the DropClass procedure. We first generate importance scores $S^1, S^2, \ldots, S^{|\mathcal{C}|}$, which are used to extract class-specific feature maps $A^1, A^2, \ldots, A^{|\mathcal{C}|}$. We then drop the class-specific feature map corresponding to a random class $z$, and average all other feature maps. Finally, $A_{\text{drop}}$ is fed into the classifier to produce the output without information of class $z$.

In this sense, dataset bias and feature entanglement are not mutually exclusive, but dataset bias is manifested as feature entanglement in segmentation tasks.

The fundamental goal of our work is to develop a training scheme that allows the model to learn debiased and disentangled representations for each class, such that the model can make robust predictions for the under-represented classes without depending on the feature representations of other classes. In essence, this is achieved by removing the class-specific features of a random class at each iteration and providing a proper loss as described in Section 2.3. Our training process not only breaks up inter-class relationships through a stochastic training scheme, but also exposes the classifier to a diverse set of class-specific feature combinations given the same input.

## 2.2 DropClass for Debiasing and Disentangling

The feature representations of multiple classes should not be entangled with one another in order to facilitate less confusion and stronger discriminability. To achieve this, we remove the feature dependencies among classes by selectively discarding the information related to a randomly chosen class at each training iteration.

The difficulty is that neural networks are black-box in nature and produce highly entangled feature representations. To alleviate this issue, we leverage Grad-CAM [7], a well-known gradient-based method of highlighting important class-specific information from entangled feature representations. Grad-CAM uses the backpropagated gradients to produce a coarse localization map that highlights important regions in the image with respect to a certain label, and is widely used to visualize class-discriminative features in deep neural networks. In similar fashion, we employ the gradients of feature $A_{i,j,k}$ with respect to the class logit $\hat{y}^c$ as the degree of importance that $A_{i,j,k}$ has on the prediction for class $c$. Since semantic segmentation is a pixel-level classification task, we average the gradients of $A_{i,j,k}$ with respect to every pixel location that corresponds to class $c$. This is used as the importance score of $A_{i,j,k}$ for class $c$, and is denoted by $S_{i,j,k}^c$:

$$S_{i,j,k}^c = \frac{1}{h \times w} \sum_{u=1}^{h} \sum_{v=1}^{w} \frac{\partial \hat{y}_{u,v,c}}{\partial A_{i,j,k}}. \tag{1}$$

The importance score for a certain class can be calculated for all locations $(i, j, k)$ of the feature map $A$ to obtain an importance score map $S^c$, with the same dimensions. Then, we perform element-wise multiplication of the feature and importance score maps to obtain class-specific feature representations, $A^c$, which is given by

$$A^c = \text{ReLU}(A \odot S^c), \tag{2}$$

where $\odot$ denotes the Hadamard product. The ReLU activation is used to preserve all features of $A$ that have a positive influence on $\hat{y}^c$ and filter out all negative ones.

With the highlighted feature representation $A^c, \forall c \in \mathcal{C}$, we are now able to discard information related to a certain class from the output of the feature extractor. In every iteration of the training

process, a class $z$ is randomly sampled as

$$z \sim U(1, |\mathcal{C}|), \tag{3}$$

where $U(1, |\mathcal{C}|)$ denotes a discrete uniform distribution across elements of $\mathcal{C}$. Through the stochastic drop of the class $z$, we deactivate $A^z$ and obtain a different combination of feature representations at each iteration, which is given by

$$A'_{\text{drop}} = \frac{1}{|\mathcal{C}|} \sum_{c=1}^{|\mathcal{C}|} \mathbb{1}[c \neq z] \cdot A^c, \tag{4}$$

where $\mathbb{1}[\cdot]$ represents an indicator function. After passing $A'_{\text{drop}}$ through a convolutional layer to compensate missing information, we obtain a new feature representation $A_{\text{drop}} = \text{conv}(A'_{\text{drop}})$ that contains information for all classes except class $z$. This feature is passed through the classifier to generate a new logit, $\hat{y}_{\text{drop}}$. Since $A_{\text{drop}}$ may still contain the information about the dropped class $z$, our approach attempts to suppress the information regarding class $z$ in the other class-specific feature maps as well, using the objective function as defined in Section 2.3. Overall, our DropClass procedure is illustrated in Figure 1.

## 2.3 Objective Function

We train the model with multiple loss functions, each with a specific goal in mind. We first evaluate the cross-entropy loss:

$$\mathcal{L}_{\text{CE}} = -\frac{1}{h \times w} \sum_{u=1}^{h} \sum_{v=1}^{w} \sum_{c=1}^{|\mathcal{C}|} y_{u,v,c} \cdot \log(\hat{p}_{u,v,c}), \tag{5}$$

where $\hat{p} = \text{softmax}(\hat{y})$ applied over the channel dimension. To account for the dropped class $z$, we design a modified cross-entropy loss that masks the pixel-wise loss at all locations, where class $z$ is the ground-truth:

$$\mathcal{L}_{\text{CE\_drop}} = -\frac{1}{h \times w} \sum_{u=1}^{h} \sum_{v=1}^{w} \sum_{c=1}^{|C|} y_{u,v,c} \cdot \log(\tilde{p}_{u,v,c}) \cdot \mathbb{1}[y_{u,v,z} \neq 1], \tag{6}$$

where $\tilde{p} = \text{softmax}(\hat{y}_{\text{drop}})$ applied over the channel dimension. Then, $\mathcal{L}_{\text{CE}}$ for the model's original output and $\mathcal{L}_{\text{CE\_drop}}$ for the DropClass output are combined by a weighted sum:

$$\mathcal{L}_{\text{seg}} = (1 - \lambda) \cdot \mathcal{L}_{\text{CE}} + \lambda \cdot \mathcal{L}_{\text{CE\_drop}}, \tag{7}$$

where $\lambda$ is a scaling constant that balances the two loss terms.

The scaling constant $\lambda$ plays an important role in ensuring the stability and effectiveness of training. The process outlined in Section 2.2 is based on the assumption that the gradient of $A_{i,j,k}$ with respect to $\hat{y}^c$ accurately quantifies the importance of $A_{i,j,k}$ on the prediction of class $c$. In the initial stages of training, however, $\hat{y}$ is likely to be largely inaccurate, and thus, the gradient may not accurately reflect the relationship between the feature and class prediction. Hence, relying on this gradient to perform DropClass from the early stages of training may lead to an unstable training process. To alleviate this issue, we train the model primarily with $\mathcal{L}_{\text{CE}}$ at the early stages of training and linearly decrease its weight as training progresses. Conversely, we gradually increase the weight for $\mathcal{L}_{\text{CE\_drop}}$, thereby facilitating learning of debiased and disentangled representations towards the end of training. This is achieved by linearly increasing the value of $\lambda$ from 0 to 1 across the duration of training. To further stabilize training, we linearly increase the probability of dropping out any class from 0 to 1 as well.

Finally, we add an additional objective term $\mathcal{L}_{\text{sup}}$ that helps suppress the output probability of the dropped class $z$:

$$\mathcal{L}_{\text{sup}} = \frac{1}{h \times w} \sum_{u=1}^{h} \sum_{v=1}^{w} \tilde{p}_{u,v,z}. \tag{8}$$

Even after removing information about class $z$, some information could still persist in the dropped feature $A_{\text{drop}}$, which would allow the model to predict a score for class $z$, albeit with low confidence. Hence, by suppressing the probability for class $z$ directly, the model is encouraged to learn

**Algorithm 1** Training scheme for DropClass.

---

**Require:** Dataset $D = \{(x(i), y(i))\}$, set of classes $\mathcal{C}$, feature extractor $g(\cdot)$ and classifier $h(\cdot)$.

1: **for** $i = 1, 2, \ldots$ **do**
2:     Compute $A = g(x(i))$, $\hat{y} = h(A)$.
3:     **for** $c = 1, 2, \ldots$ **do**
4:         Compute $S_{i,j,k}^c = \frac{1}{h \times w} \sum_{u=1}^{h} \sum_{v=1}^{w} \frac{\partial \hat{y}_{u,v,c}}{\partial A_{i,j,k}}$.             $\triangleright$ Eq. (1)
5:         Compute $A^c = \text{ReLU}(A \odot S^c)$.                    $\triangleright$ Eq. (2)
6:     **end for**
7:     Sample a class $z$ to drop and deactivate $A^z$.          $\triangleright$ Eq. (3)
8:     Gather all $A^c$ for $\forall c \in \mathcal{C} \setminus \{z\}$ and feed them into the classifier $h(\cdot)$.          $\triangleright$ Eq. (4)
9:     Compute loss function and update parameters.          $\triangleright$ Eq. (9)
10: **end for**

---

class-specific representations, *i.e.,* purge any redundant information related to class $z$ from feature representations of all other classes $c \in \mathcal{C} \setminus \{z\}$. Note that this loss is only calculated when a class is dropped. Altogether, the final objective function proposed for DropClass is:

$$\mathcal{L}_{\text{total}} = \mathcal{L}_{\text{seg}} + \alpha \mathcal{L}_{\text{sup}}, \tag{9}$$

where $\alpha$ is a hyper-parameter that weighs the relative importance of $\mathcal{L}_{\text{sup}}$. A summary of our proposed training scheme can be found in Algorithm 1.

## 3 Experiments

To evaluate the effectiveness of DropClass, we experiment on a well-known semantic segmentation dataset: Cityscapes [6] with a few reasons. First, it is relatively small, with 2975 train images. Second, it has large class imbalances, with the pixel frequency ranging from 0.1% to 36.9%. These two commonalities make it difficult for an ordinary model to learn robust, debiased representations. We also conduct experiments on the Pascal VOC dataset [14], and the results can be found in the supplementary document.

### 3.1 Implementation Details

**Network architectures** We employ two recent segmentation network architectures, HRNet [1] and DeepLabV3 [2] with a MobileNetV3 [15] backbone, to highlight the model-agnostic property of our method. For HRNet, we use the code provided by the authors[1], and use two different versions of HRNet: HRNetV1-W18 and HRNetV2-W18. For DeepLabV3, we use the official PyTorch [16] code[2] and change the default backbone to MobileNetV3. All backbones are trained on the ImageNet [17] dataset.

**Computational overhead** Calculating gradients for each class imposes a heavy computational burden, which may lead to slow training. However, we take advantage of a simple trick that allows us to calculate the gradient in Eq. (1) efficiently. Since convolution layers are linear operations, $\partial \hat{y}_{u,v,c} / \partial A_{i,j,k}$ conveniently reduces to the weight of the convolution kernel. Thus, we employ a $1 \times 1$ convolution as the final classification layer, $h$, which is a common design choice in most semantic segmentation models. Using this trick reduces both the computational and memory overhead of the gradient calculation, and allows it to be processed in parallel as well.

**Hyperparameters** We compare our model with a baseline, where the model is trained with an ordinary cross-entropy loss (Eq. (5)) only. We use the same set of hyperparameters for both experiments to ensure fair comparison. The value of the loss weighing term $\alpha$ in Eq. (9) is set to 10 for all experiments, which is based on the scale of the two loss terms. As mentioned in Section 2.3, the value of $\lambda$ is initialized as 0 and scaled linearly up to 1 until the end of training. Since DropClass depends on gradient computations to generate disentangled features, it takes more iterations for the model to fully converge under the DropClass training scheme. We sample a new class $z$ for the entire batch at

---

[1]https://github.com/HRNet/HRNet-Semantic-Segmentation
[2]https://github.com/pytorch/vision/blob/master/torchvision/models/segmentation/deeplabv3.py

Table 1: Categorical IoU scores for the Cityscapes dataset. Classes are sorted in order of increasing pixel frequency. mIoU$^\dagger$ indicates mIoU for the 9 classes with lowest pixel frequency. DeepLabV3 uses MobileNetV3 architecture as the backbone.

| | | motorcycle$^\dagger$ | rider$^\dagger$ | t.light$^\dagger$ | train$^\dagger$ | bus$^\dagger$ | truck$^\dagger$ | bicycle$^\dagger$ | t.sign$^\dagger$ | wall$^\dagger$ | fence | terrain | person | pole | sky | sidewalk | car | vegetation | building | road | mIoU | mIoU$^\dagger$ |
|---|---|---|---|---|---|---|---|---|---|---|---|---|---|---|---|---|---|---|---|---|---|---|
| Model | Pixel % | 0.1 | 0.1 | 0.2 | 0.2 | 0.2 | 0.3 | 0.4 | 0.6 | 0.7 | 0.9 | 1.2 | 1.2 | 1.2 | 4.0 | 6.1 | 7.0 | 15.9 | 22.8 | 36.9 | | |
| HRNetV1 | Baseline | 47.9 | 53.5 | 63.3 | 47.3 | 71.8 | 48.6 | 72.8 | 72.8 | 44.7 | 52.0 | 61.5 | 77.0 | 56.7 | 93.5 | 81.6 | 92.9 | 91.5 | 90.5 | 97.5 | 69.3 | 58.1 |
| | Ours | 50.2 | 55.7 | 65.3 | 48.0 | 73.4 | 48.6 | 74.1 | 75.3 | 47.1 | 53.7 | 61.9 | 78.7 | 61.6 | 94.0 | 82.3 | 93.3 | 92.1 | 91.1 | 97.6 | 70.7 | 59.7 |
| | Δ | 2.3 | 2.2 | 2.0 | 0.7 | 1.6 | 0.0 | 1.3 | 2.5 | 2.4 | 1.7 | 0.4 | 1.7 | 4.9 | 0.5 | 0.7 | 0.4 | 0.6 | 0.6 | 0.1 | 1.4 | 1.6 |
| | Δ (%) | 4.8 | 4.1 | 3.2 | 1.5 | 2.2 | 0.0 | 1.8 | 3.4 | 5.4 | 3.3 | 0.7 | 2.2 | 8.6 | 0.5 | 0.9 | 0.4 | 0.7 | 0.7 | 0.1 | 2.0 | 2.9 |
| HRNetV2 | Baseline | 58.3 | 61.5 | 69.3 | 62.7 | 80.6 | 68.2 | 76.3 | 76.3 | 50.3 | 59.5 | 65.0 | 81.0 | 62.6 | 94.1 | 84.6 | 94.4 | 92.2 | 92.0 | 98.0 | 75.1 | 67.1 |
| | Ours | 59.7 | 60.9 | 71.7 | 69.7 | 83.2 | 68.5 | 76.8 | 79.1 | 55.5 | 60.2 | 63.7 | 82.2 | 66.9 | 95.0 | 84.0 | 94.9 | 92.7 | 92.7 | 98.0 | 76.6 | 69.5 |
| | Δ | 1.4 | -0.6 | 2.4 | 7.0 | 2.6 | 0.3 | 0.5 | 2.8 | 5.2 | 0.7 | -1.3 | 1.2 | 4.3 | 0.9 | -0.6 | 0.5 | 0.5 | 0.7 | 0.0 | 1.5 | 2.4 |
| | Δ (%) | 2.4 | -1.0 | 3.5 | 11.2 | 3.2 | 0.4 | 0.7 | 3.7 | 10.3 | 1.2 | -2.0 | 1.5 | 6.9 | 1.0 | -0.7 | 0.5 | 0.5 | 0.8 | 0.0 | 2.0 | 3.8 |
| DeepLabV3 | Baseline | 45.9 | 51.6 | 53.7 | 52.3 | 66.1 | 52.3 | 68.8 | 66.0 | 46.0 | 52.9 | 57.5 | 73.1 | 48.9 | 92.6 | 79.5 | 92.1 | 90.4 | 89.7 | 97.4 | 67.2 | 55.9 |
| | Ours | 49.7 | 53.1 | 53.7 | 55.0 | 67.4 | 51.9 | 68.4 | 66.8 | 47.1 | 52.4 | 58.0 | 73.5 | 48.5 | 92.8 | 79.1 | 92.3 | 90.3 | 89.9 | 97.3 | 67.7 | 57.0 |
| | Δ | 3.8 | 1.5 | 0.0 | 2.7 | 1.3 | -0.4 | -0.4 | 0.8 | 1.1 | -0.5 | 0.5 | 0.4 | -0.4 | 0.2 | -0.4 | 0.2 | -0.1 | 0.2 | -0.1 | 0.5 | 1.1 |
| | Δ (%) | 8.3 | 2.9 | 0.0 | 5.2 | 2.0 | -0.8 | -0.6 | 1.2 | 2.4 | -0.9 | 0.9 | 0.5 | -0.8 | 0.2 | -0.5 | 0.2 | -0.1 | 0.2 | -0.1 | 0.7 | 2.3 |

each training iteration. The rest of our hyperparameters are organized as a table in the supplementary materials, where we detail the number of iterations, batch size, learning rate, learning rate decay, image size, and type of GPU used for each of our experiments.

## 3.2 Quantitative and Qualitative Results

Tables 1 presents the categorical Intersection-over-Union (IoU) and mean-IoU (mIoU) scores on the Cityscapes dataset. The absolute change in IoU ($\Delta$), as well as the relative change in IoU ($\Delta(\%)$) are reported, to compare our method with the baseline. For better interpretability of our results, the classes are sorted in order of increasing pixel frequency, and we also calculate a separate mIoU score for the least frequent 50% of classes, which is denoted by mIoU$^\dagger$.

Our method consistently outperforms the baseline across all models, with 2.0%, 2.0%, and 0.7% gains on the mIoU for HRNetV1, HRNetV2, and DeepLabV3 models, respectively. More importantly, we observe even larger improvements on the mIoU for the 9 most under-represented classes, which account for merely 2.8% of all pixels: 2.9% on HRNetV1, 3.8% on HRNetV2, and 2.3% on DeepLabV3. These empirical results supports our claim that DropClass helps the model learn debiased feature representations that benefit the model's overall performance, especially for under-represented classes. There are a few more interesting details we observe from Table 1. First, the IoU improves on most classes, albeit with smaller margins on well-represented classes. Also, we notice especially large improvements across all models on the "train" and "wall" categories, which also happen to be two of the least frequently appearing categories: "train" is the least frequent class, appearing in only 0.4% of all images, and "wall" is the fifth least frequent class, appearing in only 2.8% of all images. Figure 2 illustrates the qualitative results of the proposed model in comparison to the baseline on the Cityscapes dataset.

## 3.3 Comparisons with Other Debiasing Techniques

Many works on debiasing focus on the image classification task, and are not easily applicable to the semantic segmentation task. To emphasize that DropClass is better suited for semantic segmentation, we provide comparisons with two baseline debiasing techniques: reweighting and resampling.

**Reweighting** In Table 1, the results of both the baseline and our model employ median frequency balancing, which is widely used for the Cityscapes dataset. In median frequency balancing, the cross-entropy term is reweighted according to the frequency of classes. We train the model without this reweighting scheme on both the baseline and the model trained with DropClass. As shown in Table 2, even without the reweighting scheme, the model trained with DropClass outperforms the baseline model with the reweighting scheme.

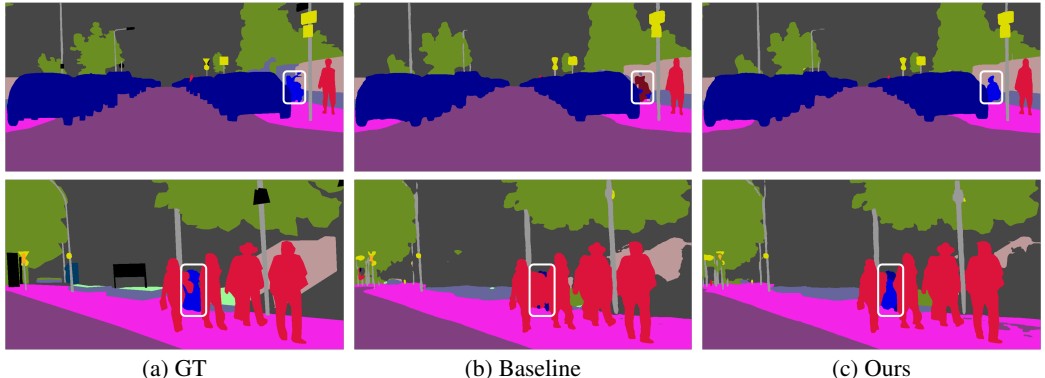

|            (a) GT            |          (b) Baseline          |          (c) Ours          |

Figure 2: Visualization of segmentation results on Citycapes. Due to the semantic similarity or spatial co-occurrence between classes, the baseline model misclassifies the least frequent class "motorcycle" as "bicycle" (top) or "person" (bottom), while the model trained with DropClass predicts correctly for both cases. The white boxes in the images denote the regions with notable differences between the two models.

Table 2: Comparison with the reweighting baselines on HRNetV1.

|                          | mIoU | mIoU$^\dagger$ |
| ------------------------ | ---- | -------------- |
| Baseline w/o reweighting | 69.2 | 57.0           |
| Baseline                 | 69.3 | 58.1           |
| Ours w/o reweighting     | 70.1 | 58.6           |
| Ours                     | **70.7** | **59.7**   |

Table 3: Comparison with "motorcycle" resampling on HRNetV1 without reweighting.

|                            | IoU (motor.) | mIoU |
| -------------------------- | ------------ | ---- |
| Baseline                   | 39.9         | 69.2 |
| Baseline + 2x "motorcycle" | 46.9         | 69.1 |
| Ours                       | **49.7**     | **70.1** |

**Resampling**   Resampling is not straightforward to implement in semantic segmentation since the training images contain multiple classes. For this experiment, we focus solely on the class with least pixel frequency of Cityscapes ("motorcycle"), without any reweighting scheme. To increase the sampling frequency of the "motorcycle" class, we enlarge the training set such that each image containing the "motorcycle" class is sampled twice at every training epoch. We observe a +7%p increase of IoU ($39.9\% \rightarrow 46.9\%$) in the baseline by resampling the "motorcycle" class. However, the baseline with "motorcycle" resampling loses mIoU by 0.1%p, meaning that the improvement of the "motorcycle" class comes at the cost of performance on other classes. To the contrary, our model outperforms the baseline with "motorcycle" resampling in terms of both the metrics, by +2.8%p, +1.0%p, respectively. The results are shown in Table 3.

## 3.4   Analysis

**Robustness to harmful bias effects**   To demonstrate the debiasing effects of DropClass, we conduct a simple experiment to analyze the co-location effects between classes. Since it is difficult to rigorously quantify the co-location and analyze the bias vulnerability of the classes in real-world segmentation datasets, we define a simple proxy instead: the drop in IoU for each class pair (e.g., IoU of "person" after erasing "road") of the baseline model. To achieve this, we design an unbiased test set (I*), where the validation set is copied 19 times, and one of the 19 classes in Cityscapes is erased from each copy. By observing how the IoU of class Y is affected by masking out the pixels of class X, we can quantify the harmful effects of co-location. For example, the "person" class is most affected by the absence of "road", "sidewalk" and "bicycle" classes, which suggests that these classes are highly co-located with the "person" class. For each evaluated class, we first identify the Top3 most influential classes, then average IoUs after removing those classes. The same sets of Top3 classes are used to evaluate the model trained with DropClass on the unbiased test set. The results are shown in Table 4, where the baseline model obtains the mIoU of 68.9% while our model achieves the mIoU of 70.9%. Across the majority classes, we observe that our model suffers from less performance degradation, which verifies that the model trained with DropClass is less susceptible to the negative effects of co-location during inference. Also, note that the "rider" class, by definition, is dependent on the presence of "motorcycle" or "bicycle" class, which may rationalize the reason behind the smaller difference between the baseline and our model.

Table 4: Quantifying the effects of co-location between classes with respect to bias. Top3 most influential classes are selected from the baseline model. mIoU* indicates mIoU for the 9 classes with the highest bias vulnerability.

| | rider | motorcycle | train | wall | t.light | bicycle | bus | person | pole | sidewalk | fence | terrain | truck | sky | car | vegetation | building | t.sign | road | mIoU | mIoU* |
|---|---|---|---|---|---|---|---|---|---|---|---|---|---|---|---|---|---|---|---|---|---|
| Baseline (I) | 62.6 | 60.5 | 67.8 | 54.9 | 71.0 | 76.5 | 87.5 | 81.7 | 63.3 | 85.9 | 62.6 | 66.2 | 73.2 | 94.3 | 94.4 | 91.6 | 92.2 | 77.7 | 98.2 | 77.0 | 69.5 |
| Baseline (I*) | 33.0 | 37.8 | 53.9 | 40.5 | 59.2 | 66.5 | 80.3 | 74.3 | 60.6 | 79.3 | 57.7 | 60.1 | 68.6 | 91.8 | 92.2 | 89.4 | 90.0 | 76.7 | 97.9 | 68.9 | 56.2 |
| Δ (Baseline) | -29.6 | -22.7 | -13.9 | -14.4 | -11.8 | -10.0 | -7.3 | -7.5 | -2.7 | -6.6 | -4.9 | -6.2 | -4.6 | -2.6 | -2.2 | -2.2 | -2.1 | -1.1 | -0.3 | -8.0 | -13.3 |
| Ours (I) | 61.7 | 61.9 | 72.6 | 59.1 | 72.0 | 76.9 | 89.2 | 82.5 | 67.8 | 85.5 | 63.1 | 64.8 | 72.9 | 95.2 | 94.8 | 92.0 | 92.7 | 79.1 | 98.1 | 78.0 | 71.5 |
| Ours (I*) | 34.0 | 43.8 | 61.0 | 45.8 | 62.1 | 67.1 | 82.6 | 76.3 | 65.2 | 83.0 | 57.1 | 59.1 | 66.8 | 92.4 | 93.5 | 90.3 | 90.4 | 77.9 | 97.9 | 70.9 | 59.8 |
| Δ (Ours) | -27.7 | -18.1 | -11.6 | -13.3 | -9.8 | -9.8 | -6.6 | -6.2 | -2.7 | -2.5 | -6.0 | -5.7 | -6.0 | -2.7 | -1.4 | -1.6 | -2.3 | -1.2 | -0.2 | -7.1 | -11.7 |

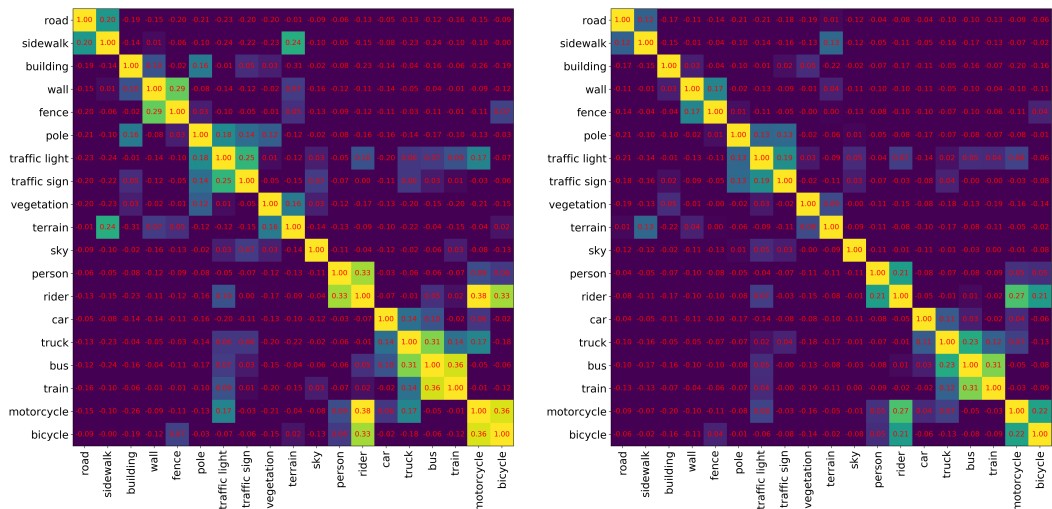

Figure 3: Correlation matrices of the classifier weights for the baseline model (left) and the model trained with DropClass (right). The inter-class similarity of our model is clearly smaller than that of the baseline model, which implies that DropClass is effective to eliminate the correlation between the representations of similar classes.

**Correlation analysis**   We find that the baseline model is not only prone to learning biases due to the co-location of classes, but also susceptible to learning correlated representations, especially on under-represented classes. Thus, to better illustrate the benefits of DropClass, we plot the correlation matrix of both models, where the element at $(i, j)$ corresponds to the cosine similarity between weight vectors of the i-th and j-th clasees in the final classification layer. We notice that the most frequent "road" class has low similarities with all classes, while the least frequent "motorcycle" class has high similarities with multiple classes, such as "bicycle" and "rider". To this end, we calculate the row-wise non-diagonal sum, which is a measure of how correlated a class representation is with respect to all other classes and report the numbers in Table 5. We observe that more frequently appearing classes tend to be less correlated with other classes. Note that there is a clear trend between increasing frequency and decreasing correlations among classes. Furthermore, we observe that the model trained with DropClass shows lower inter-class correlations for all class pairs, as summarized in the last row of Table 5. The full correlation matrix is shown at Figure 3.

**Grad-CAM visualizations**   We further analyze our method by examining Grad-CAM visualizations of trained models. We first select three pairs of classes that are likely to confuse a segmentation model: "motorcycle ↔ bicycle", "truck ↔ train", and 'bus ↔ truck". Then we generate the Grad-CAM visualizations with the baseline and the model trained with DropClass (HRNetV1 on Cityscapes), and compare the results in Figure 4, where each row corresponds to a different class pair. The second and third columns correspond to the visualizations of the two models when visualizing the Grad-CAMs for the ground-truth (GT) classes while the next two columns present the Grad-CAMs with respect to non-GT classes. Compared to the visualizations of the baseline model, the visualizations of our model

Table 5: Class correlations. Classes are sorted in order of increasing pixel frequency. $\Delta$ indicates correlations of the baseline model subtracted by correlations of the model trained with DropClass.

| | motorcycle† | rider† | t.light† | train† | bus† | truck† | bicycle† | t.sign† | wall† | fence | terrain | person | pole | sky | sidewalk | car | vegetation | building | road |
|---|---|---|---|---|---|---|---|---|---|---|---|---|---|---|---|---|---|---|---|
| Baseline | 1.24 | 1.20 | 0.97 | 0.65 | 0.91 | 0.88 | 0.85 | 0.60 | 0.46 | 0.45 | 0.53 | 0.48 | 0.63 | 0.16 | 0.45 | 0.30 | 0.35 | 0.33 | 0.20 |
| Ours | 0.74 | 0.77 | 0.65 | 0.47 | 0.63 | 0.58 | 0.52 | 0.39 | 0.24 | 0.23 | 0.26 | 0.31 | 0.28 | 0.08 | 0.25 | 0.19 | 0.16 | 0.09 | 0.12 |
| $\Delta$ | 0.50 | 0.43 | 0.32 | 0.18 | 0.28 | 0.30 | 0.33 | 0.21 | 0.22 | 0.22 | 0.27 | 0.17 | 0.35 | 0.08 | 0.20 | 0.11 | 0.19 | 0.24 | 0.08 |

| Input | Baseline: GT | Ours: GT | Baseline: non-GT | Ours: non-GT |

Figure 4: Examples of Grad-CAM visualizations. From top to bottom, the input images contain "motorcycle", "truck", and "bus". The Grad-CAMs of the baseline and the model trained with DropClass with respect to the ground-truth (GT) classes are presented in the second and third columns respectively, while the Grad-CAMs of the two models with respect to the non-GT classes—"bicycle", "train", and "truck"—are shown in the fourth and fifth columns.

consistently display lower activations for the non-GT classes. Furthermore, our model exhibits strong gradients for the GT classes as well, which indicates that the model maintains better discriminability in the process of debiasing.

**Ablations** To better understand our proposed method and its effectiveness, we conduct ablation experiments on the Cityscapes dataset with the HRNetV1 model. First, we omit $\mathcal{L}_{\text{sup}}$ defined in Eq (8), which has the purpose of suppressing outputs for the dropped class. We observe that when we remove this loss term, the effectiveness of our method decreases slightly in terms of the total mIoU and the under-represented class mIoU (mIoU†), by 0.5%p and 1.1%p, respectively. However, it still outperforms the baseline by 0.9%p and 0.5%p on the two metrics. Next, we also test the effects of dropping the labels of a randomly selected class, without DropClass or any of its accompanying loss functions. In other words, a model is trained only with the ordinary cross-entropy loss, Eq. (5), while random class labels are not used (e.g., ignored) at each iteration. Unsurprisingly, we observe that this model performs worse than the baseline

Table 6: Ablation experiments on Cityscapes with HRNetV1.

| Method | mIoU | mIoU† |
|---|---|---|
| Baseline | 69.3 | 58.1 |
| Ours | 70.7 | 59.7 |
| $\Delta$ | 1.4 | 1.6 |
| $\Delta$ (%) | 2.0 | 2.9 |
| w/o Eq. (8) | 70.2 | 58.6 |
| $\Delta$ | 0.9 | 0.5 |
| $\Delta$ (%) | 1.4 | 0.9 |
| Ignore Label | 69.2 | 58.0 |
| $\Delta$ | -0.1 | -0.1 |
| $\Delta$ (%) | -0.1 | -0.1 |

model, by 0.1%p in both metrics. Thus, our ablation experiments highlight the importance of the suppression loss and validate the effectiveness of DropClass.

# 4  Related Works

**Dataset bias**   Dataset bias is a critical issue in modern machine learning because trained models often achieve outstanding accuracy simply by capturing the correlated features instead of identifying the proper representations of target classes. To tackle the limitation, various debiasing techniques have been proposed under supervised [19–22] and unsupervised [13, 23, 24] settings, but most of existing approaches are for simple classification tasks. Technically, they are limited to sample reweighting [24–27] and loss adjustment [19–24]. The proposed algorithm is unique in the sense that we aim to learn a classifier for dense prediction. One previous work [28] argues that feature entanglement in the early layers can enhance the discriminative power of the model. However, they experiment on the classification task, and the conclusion may not be applicable to semantic segmentation, where data samples contain multiple classes per image. We suspect that explicitly entangling features in a segmentation model may have an adverse effect on the model's ability to identify class boundaries within a given image.

**Semantic segmentation**   Semantic segmentation works have attempted to improve performance with diverse operations that aim to widen the receptive field, while maintaining details in the features from different resolutions. To this end, DeconvNet [3] learns deconvolution layers to replace fixed upsampling operations and DeepLabV3 [2] employs atrous convolution to capture objects at multiple scales. Furthermore, PSPNet [29] proposes a pyramid pooling module to capture different sub-region representations and HRNet [1] aggregates four different resolution of feature maps to preserve details, especially for the deeper layers with lower resolutions.

**Visual explanations**   Even with extensive results in broad domains, non-convexity and high-dimensionality of deep neural networks restrict human-level understandings of its internal process. To remedy this issue, [7, 30–32] attempt to make visual explanations of deep models. CAM [30] first proposes to find discriminative regions relevant to target classes, but has architectural restriction due to the use of the global average pooling layer. Grad-CAM [7] visualizes the importance of each class by leveraging the gradient information, and [31] extends Grad-CAM for semantic segmentation. CAM and Grad-CAM have often been employed in weakly supervised object detection [33, 34] and segmentation [35–37], where they highlight relevant objects or regions without full supervision for target tasks. We compute Grad-CAM for each class, but employ the estimated class-specific feature maps for learning debiased and disentangled representations.

# 5  Conclusion and Discussion

We presented a model agnostic training scheme that helps the model learn more debiased and disentangled feature representations for the semantic segmentation task. Our method is based on the simple and intuitive idea of randomly dropping disentangled class-specific features at each iteration of training. The new combination of features removes certain inter-class dependencies, which leads to learning debiased and disentangled feature representations. Our experimental results on two distinct network architectures and datasets validate our method's effectiveness, and our analysis reflect that our method achieves the intended goals.

One limitation of the proposed DropClass training scheme is that it requires more iterations for the model to fully converge. This is because DropClass depends on the model's ability to generate disentangled feature representations and has more complex training process.

**Acknowledgments**

We truly thank the anonymous reviewers for their valuable comments throughout the review process. This paper was improved significantly through the communication with them. This work was partly supported by Samsung Advanced Institute of Technology, Korean ICT R&D programs of the MSIT/IITP grant [2017-0-01779, XAI, 2021-0-01343, Artificial Intelligence Graduate School Program (Seoul National University)], and the Bio & Medical Technology Development Program of the National Research Foundation (NRF) funded by the Korea government (MSIT) [2021M3A9E4080782].

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
