# Learning Debiased and Disentangled Representations for Semantic Segmentation
## *Supplementary Document*

**Sanghyeok Chu**      **Dongwan Kim**      **Bohyung Han**
ECE & ASRI, Seoul National University
`{sanghyeok.chu,dongwan123,bhhan}@snu.ac.kr`

## A   Verification of Feature Disentanglement

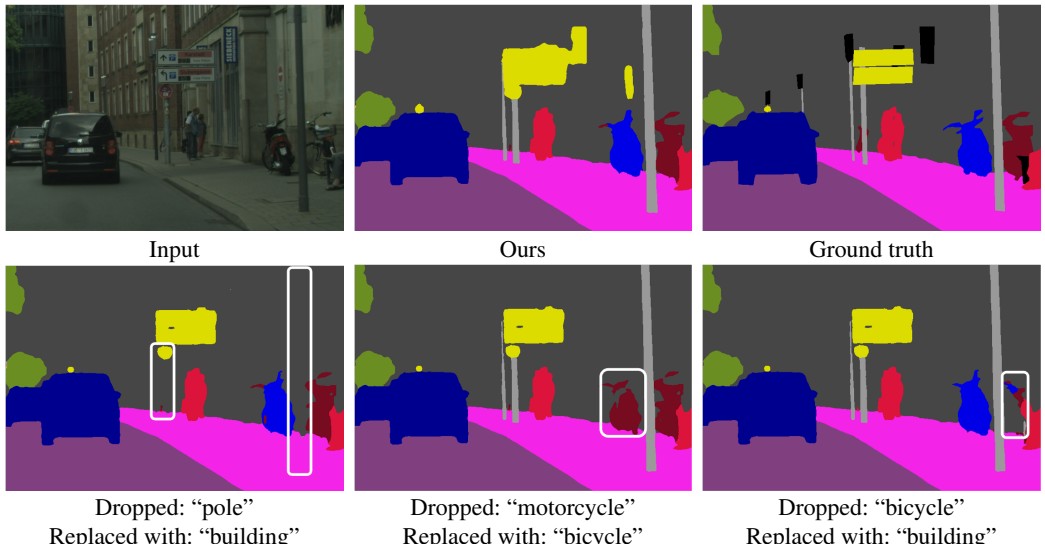

Figure A-1: Reconstruction results for the dropped classes. (Top) Segmentation results from the proposed method (Bottom) Results after dropping classes, "pole" (gray), "motorcycle" (blue) and "bicycle" (brown). The relevant regions are marked with rectangles.

To verify whether the proposed method successfully leads disentangled representation learning or not, we conduct a simple experiment to check reconstructability of a dropped class. Through this experiment, our goal is to confirm a following hypothesis: If the model has learned disentangled representations, the gradients would truly propagate class-specific information and enable class drop in the feature representation. In other words, there would be less redundancy between feature representations among different classes.

We first make a new randomly initialized classifier $h'(\cdot)$ and fix a predetermined class $z$ to drop such that class $z$ is dropped at every iteration. Then we extract $A_{\text{drop}}$, which contains all information except for the dropped class $z$, and pass it to $h'(\cdot)$ to obtain a new prediction $\hat{y}_{\text{drop}}$. With that prediction, $h'(\cdot)$ is trained on Cityscapes with all labels, including the dropped class $z$. If the model has truly learned disentangled representations, it should not be able to reconstruct predictions for class $z$ since the related features have been dropped from the extracted features. Otherwise, it should be able to reconstruct predictions for class $z$ using the feature representations of other classes.

35th Conference on Neural Information Processing Systems (NeurIPS 2021).

Table B-1: Quantifying the effects of bias. Top3 most influential classes are selected from the DropClass model. mIoU* indicates mIoU for the 9 classes with the highest bias vulnerability.

| | rider | motorcycle | wall | train | bicycle | t.light | truck | bus | fence | person | terrain | sky | pole | sidewalk | building | vegetation | car | t.sign | road | mIoU | mIoU* |
|---|---|---|---|---|---|---|---|---|---|---|---|---|---|---|---|---|---|---|---|---|---|
| Baseline (I) | 62.6 | 60.5 | 54.9 | 67.8 | 76.5 | 71.0 | 73.3 | 87.5 | 62.6 | 81.7 | 66.2 | 94.3 | 63.3 | 85.9 | 92.2 | 92.4 | 94.4 | 77.7 | 98.2 | 77.0 | 68.5 |
| Baseline (I*) | 33.0 | 37.8 | 40.5 | 54.2 | 66.5 | 59.2 | 69.0 | 80.3 | 58.0 | 74.3 | 60.1 | 91.8 | 60.6 | 79.4 | 90.0 | 90.3 | 92.2 | 76.7 | 97.9 | 69.0 | 55.4 |
| Δ (Baseline) | -29.6 | -22.7 | -14.4 | -13.5 | -10.0 | -11.8 | -4.3 | -7.3 | -4.6 | -7.5 | -6.2 | -2.6 | -2.7 | -6.6 | -2.1 | -2.2 | -2.2 | -1.1 | -0.3 | -8.0 | -13.1 |
| Ours (I) | 61.7 | 61.9 | 59.1 | 72.6 | 76.9 | 72.0 | 72.7 | 89.2 | 63.2 | 82.5 | 64.8 | 95.2 | 67.8 | 85.6 | 92.7 | 92.7 | 94.8 | 79.1 | 98.1 | 78.0 | 69.9 |
| Ours (I*) | 34.0 | 43.8 | 45.8 | 60.0 | 67.1 | 62.1 | 65.3 | 82.6 | 56.8 | 76.3 | 59.1 | 92.4 | 65.2 | 83.0 | 90.4 | 91.0 | 93.5 | 77.9 | 97.9 | 70.8 | 57.5 |
| Δ (Ours) | -27.7 | -18.1 | -13.3 | -12.6 | -9.8 | -9.8 | -7.4 | -6.6 | -6.4 | -6.2 | -5.7 | -2.7 | -2.7 | -2.6 | -2.3 | -1.8 | -1.4 | -1.2 | -0.2 | -7.3 | -12.4 |

The qualitative results shown in Figure A-1 support the aforementioned hypothesis. When the dropped classes are "pole", "motorcycle", and "bicycle", the HRNetV1 model trained with DropClass is unable to generate confident predictions for the dropped classes; instead, it makes predictions for other classes.

# B  Further Analysis about Co-location and Bias Vulnerability

**Quantitative analysis**   In Section 3.4 of our main paper, we present an analysis on the robustness to harmful bias effects with Top3 most influential classes, which are selected from the baseline model. In the interest of fairness, we further present the results for the same experiment, except that the Top3 class selection scheme is based on the model trained with DropClass in Table B-1. In this way, we expect the drop in IoU to be larger in the our model and smaller in the baseline model when each of them are compared to the results with Top3 class selection scheme based on the baseline model .

As seen in Table B-1, the baseline model obtains an mIoU of 69.0%, while our model obtains an mIoU of 70.8%. As expected, the mIoUs of the models are changed by +0.1%p (68.9% → 69.0%) and -0.1%p (70.9% → 70.8%) for each the baseline and our model, which verifies the fairness in our experimental settings. Still, we observe the same tendency that the model trained with DropClass suffers from less performance degradation across the majority of the classes.

**Qualitative analysis**   We further provide a qualitative analysis on the robustness to harmful bias effects. Since the "road" class is selected as one of the Top3 most influential classes for the majority of other classes, we construct unbiased test images (I*) by erasing all pixels of "road" in the original image (I). We visualize segmentation outputs for both images (I, I*) and mark notable regions with white boxes in Figure B-2.

As seen in Figure B-2, the predictions of the baseline model on the "sidewalk" class are changed into other classes, such as "building", "wall", and "terrain" in the prediction for the first test case. More interestingly, we observe two significant changes in the baseline models' predictions for the second test case; predictions on "bicycle" class is vanished in the absence of the "road" class, which further leads to change predictions on "rider" into "person" class. In contrast, the model trained with DropClass less suffers from the absence of "road" class and preserve its original predictions on both test cases.

# C  Quantitative Results on Pascal VOC

We additionally present a quantitative results with Pascal VOC [4] dataset. There are a few key differences between Pascal VOC and Cityscapes [2] datasets that help illustrate our method's efficacy in diverse settings. First, Cityscapes is an urban road scene dataset where images are captured from the dashcam of a vehicle, while the images in Pascal VOC are more random images that are captured in every-day life. Second, the average number of classes that appear in each image vary greatly between the two datasets: while Cityscapes has an average of around 12 classes per image, Pascal VOC [4] averages at 1.5 classes per image (without the "background" class). All in all, the two datasets provide a diverse and challenging environment to demonstrate the effectiveness of our proposed method.

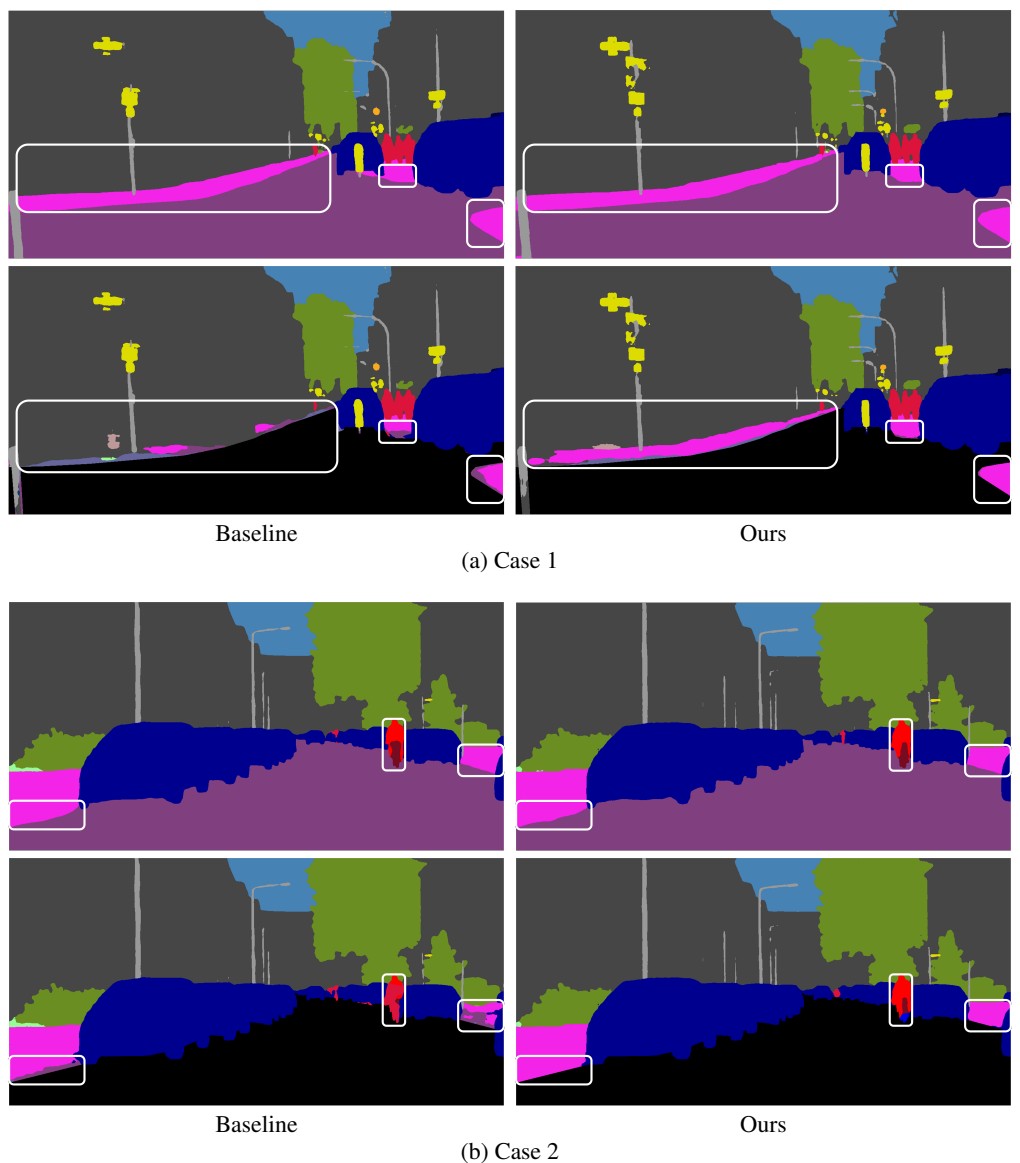

| Baseline | Ours |

(a) Case 1

| Baseline | Ours |

(b) Case 2

Figure B-2: Visualization of the robustness to harmful bias effects. For both cases, the top row denotes the output of each model on the original test image, while the bottom row denotes the output of each model on an unbiased test image, which is generated by erasing the "road" class from the original test image. The boxes highlight the regions with notable changes in each image.

In Pascal VOC, all pixels that do not belong to one of the 20 specified classes are annotated as the "background" class. Due to the ambiguity regarding the "background" class, it may be inherently difficult for the model to learn class-specific information for this class, and may also have adverse effects in our DropClass method since we aim to drop class-specific features. Therefore, we conduct experiments on Pascal VOC without the "background" class in Table C-2. For better interpretability of our results, the classes are sorted in order of increasing frequency, and we also calculate a separate mIoU score for the least frequent 50% of classes (denoted by mIoU$^{\dagger}$). However, it is important to note that we employ a different metric to define frequency from Cityscapes. On Cityscapes, we count the number of pixels that appear in the training set for each class, and normalize the counts to obtain the pixel frequency. We use this metric because each image in Cityscapes contains a large number of classes, and the average size of each class varies greatly, e.g., one instance of "road" has far more pixels than one instance of "pole". On Pascal VOC, however, only a few classes appear per image (1.5), and the class sizes do not vary as much as they do in Cityscapes. Thus, we count the number of images that each class appears in, and calculate the normalized instance frequency.

Table C-2: Categorical IoU scores for the Pascal VOC segmentation dataset. Classes are sorted in the order of increasing instance frequency. mIoU† indicates mIoU for the 10 classes with the lowest instance frequency. DeepLabV3 uses MobileNetV3 architecture as the backbone.

| | | cow† | sheep† | bus† | pottedplant† | bicycle† | horse† | boat† | motorbike† | sofa† | train† | diningtable | tvmonitor | bottle | aeroplane | bird | car | cat | chair | dog | person | mIoU | mIoU† |
|---|---|---|---|---|---|---|---|---|---|---|---|---|---|---|---|---|---|---|---|---|---|---|---|
| | Instance % | 1.7 | 1.9 | 2.4 | 2.7 | 2.8 | 2.8 | 2.8 | 3.0 | 3.0 | 3.1 | 3.2 | 3.2 | 3.5 | 3.6 | 4.2 | 5.6 | 5.8 | 5.9 | 6.6 | 14.5 | | |
| HRNetV1 | Baseline | 50.4 | 59.3 | 83.0 | 73.4 | 60.8 | 56.6 | 66.9 | 71.0 | 49.0 | 81.6 | 60.2 | 74.7 | 62.0 | 82.4 | 68.8 | 80.5 | 77.3 | 33.8 | 61.5 | 82.4 | 66.8 | 65.2 |
| | Ours | 52.4 | 66.9 | 79.0 | 72.8 | 62.8 | 53.3 | 65.4 | 75.9 | 44.8 | 79.2 | 57.0 | 78.1 | 64.2 | 84.4 | 70.8 | 83.2 | 79.2 | 31.7 | 64.4 | 81.1 | 67.3 | 65.3 |
| | Δ | 2.0 | 7.6 | -4.0 | -0.6 | 2.0 | -3.3 | -1.5 | 4.9 | -4.2 | -2.4 | -3.2 | 3.4 | 2.2 | 2.0 | 2.0 | 2.7 | 1.9 | -2.1 | 2.9 | -1.3 | 0.5 | 0.1 |
| | Δ (%) | 4.0 | 12.8 | -4.8 | -0.8 | 3.3 | -5.8 | -2.2 | 6.9 | -8.6 | -2.9 | -5.3 | 4.6 | 3.5 | 2.4 | 2.9 | 3.4 | 2.5 | -6.2 | 4.7 | -1.6 | 0.7 | 0.2 |
| HRNetV2 | Baseline | 61.2 | 74.4 | 91.7 | 79.6 | 74.8 | 65.7 | 80.9 | 85.6 | 58.5 | 92.1 | 72.0 | 87.2 | 78.7 | 92.2 | 83.6 | 87.5 | 89.6 | 45.9 | 77.5 | 87.4 | 78.3 | 76.5 |
| | Ours | 70.5 | 77.2 | 91.4 | 84.1 | 74.0 | 70.5 | 87.5 | 87.7 | 58.4 | 90.4 | 70.8 | 86.4 | 78.1 | 92.5 | 85.5 | 89.7 | 90.2 | 46.0 | 78.6 | 87.6 | 79.9 | 79.2 |
| | Δ | 9.3 | 2.8 | -0.3 | 4.5 | -0.8 | 4.8 | 6.6 | 2.1 | -0.1 | -1.7 | -1.2 | -0.8 | -0.6 | 0.3 | 1.9 | 2.2 | 0.6 | 0.1 | 1.1 | 0.2 | 1.6 | 2.7 |
| | Δ (%) | 15.2 | 3.8 | -0.3 | 5.7 | -1.1 | 7.3 | 8.2 | 2.5 | -0.2 | -1.8 | -1.7 | -0.9 | -0.8 | 0.3 | 2.3 | 2.5 | 0.7 | 0.2 | 1.4 | 0.2 | 2.0 | 3.6 |
| DeepLabV3 | Baseline | 77.4 | 87.4 | 96.9 | 81.1 | 80.5 | 76.3 | 87.6 | 84.7 | 48.7 | 97.2 | 68.3 | 85.3 | 83.1 | 94.5 | 89.9 | 88.9 | 90.3 | 42.2 | 82.2 | 85.3 | 81.4 | 81.8 |
| | Ours | 76.9 | 89.1 | 96.4 | 77.6 | 81.9 | 78.2 | 85.6 | 84.6 | 53.3 | 97.2 | 65.5 | 86.4 | 81.9 | 92.9 | 89.6 | 88.5 | 91.8 | 45.5 | 83.5 | 86.0 | 81.6 | 82.1 |
| | Δ | -0.5 | 1.7 | -0.5 | -3.5 | 1.4 | 1.9 | -2.0 | -0.1 | 4.6 | 0.0 | -2.8 | 1.1 | -1.2 | -1.6 | -0.3 | -0.4 | 1.5 | 3.3 | 1.3 | 0.7 | 0.2 | 0.3 |
| | Δ (%) | -0.6 | 2.0 | -0.6 | -4.3 | 1.8 | 2.4 | -2.3 | -0.1 | 9.4 | 0.0 | -4.2 | 1.3 | -1.4 | -1.7 | -0.3 | -0.5 | 1.7 | 7.9 | 1.6 | 0.8 | 0.3 | 0.3 |

Table E-3: Batch sizes and learning rates of our experiments.

| Dataset | Model | Batch size | Learning rate |
|---|---|---|---|
| | HRNetV1 | 48 | 0.04 |
| Cityscapes | HRNetV2 | 48 | 0.04 |
| | DeeplabV3 | 12 | 0.01 |
| | HRNetV1 | 64 | 0.016 |
| Pascal VOC | HRNetV2 | 64 | 0.016 |
| | DeeplabV3 | 48 | 0.012 |

Our model outperforms the overall mIoU by 0.7%, 2.0%, and 0.3% on HRNetV1, HRNetV2, and DeepLabV3, respectively, which demonstrates that our method is effective on a wide range of datasets, regardless of their characteristics. Compared to the results on Cityscapes, however, the performance gains are much less consistent across all classes. In other words, we observe categorical performance drops more frequently. We hypothesize that this behavior is caused by the low number of average classes per image in Pascal VOC (1.5 classes per image without the "background" class, compared to 12 classes per image on Cityscapes). Since fewer classes appear in each image, there is inherently less bias caused by inter-class relationship that our model can reduce. Thus, the IoU gains are less consistent on this dataset than they are on the Cityscapes dataset.

# D   Qualitative Results on Cityscapes

Figure D-3 shows more qualitative results on the Cityscapes dataset in addition to the ones presented in the main paper.

# E   Implementation Details

We mostly follow the default configurations provided by the official HRNet [1] code[1] for the Cityscapes [2]. Similarly, we refer to the default configurations of the Pascal Context [3] of the HRNet code for Pascal VOC [4]. However, for both datasets, we increase batch size, and also adjust the learning rates by applying linear scaling rule [5] to accommodate for the increase in batch size. The batch size and learning rate settings for each experiment is organized in Table E-3. The settings listed in Table E-3 apply for both the baseline and the model trained with DropClass. The baseline model is trained for 484 and 200 epochs for Cityscapes and Pascal VOC, respectively, which is given by a default configuration in the official HRNet [1] code. For DropClass, we train the model for 968 and 400 epochs for Cityscapes and Pascal VOC for each datasets. Furthermore, for the data augmentation, we perform random horizontal flipping, random scaling in the range of [0.5, 2],

---

[1]`https://github.com/HRNet/HRNet-Semantic-Segmentation`

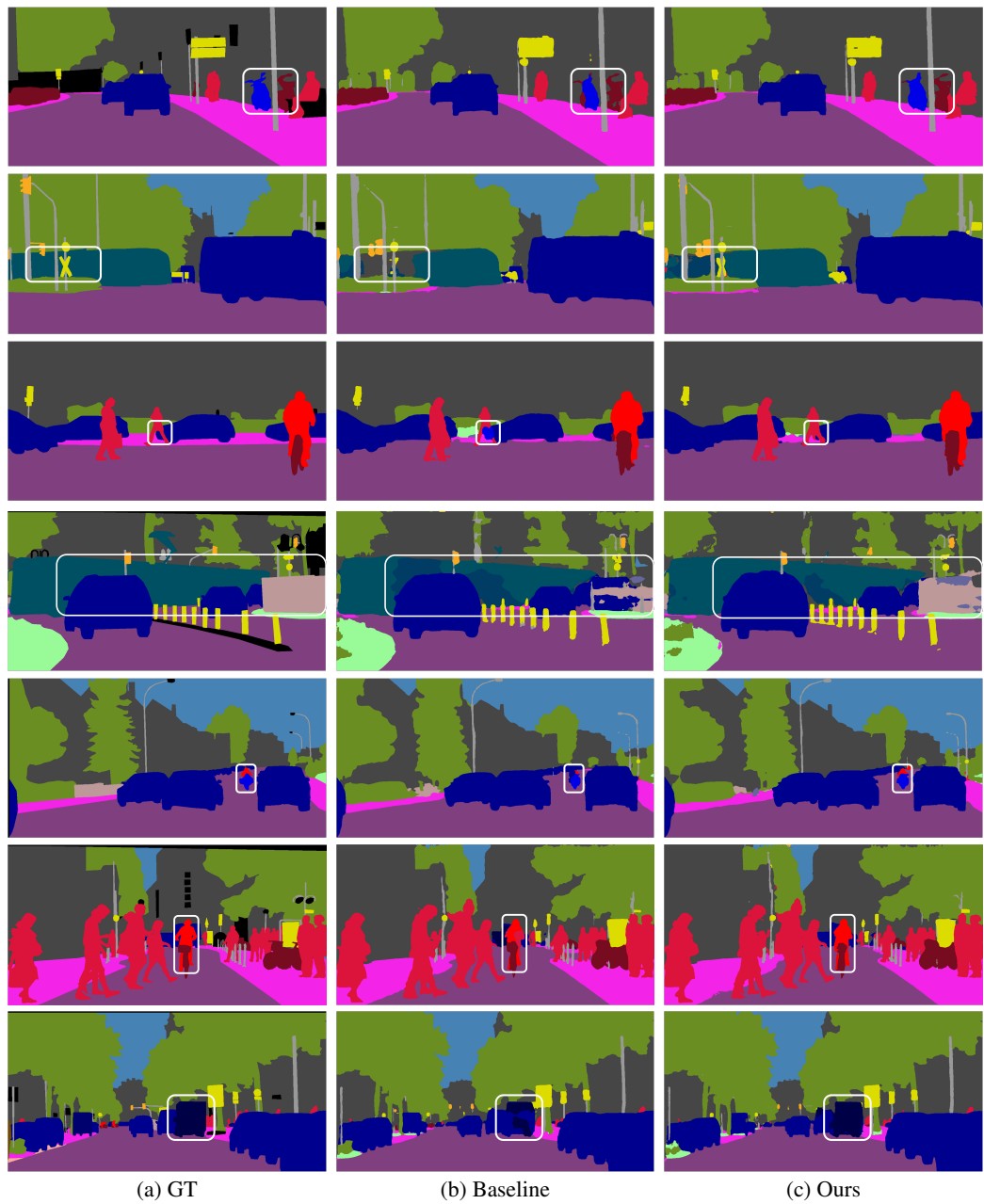

| (a) GT | (b) Baseline | (c) Ours |

Figure D-3: Visualization of segmentation results on the Citycapes dataset. From left to right, we present ground-truth, predictions of the baseline model, and predictions of the model trained with DropClass. Regions with notable differences between two models are marked with rectangles.

and random cropping of size 512 × 1024 for Cityscapes and 480 × 480 for Pascal VOC. For all experiments, we perform the polynomial learning rate policy. Finally, all experiments are conducted on four Quadro RTX 8000 GPUs.