# OpenReview forum: "Learning Debiased and Disentangled Representations for Semantic Segmentation"
_NeurIPS.cc/2021/Conference — NeurIPS 2021 Poster_

### Official Review · Reviewer_K1Gz · 2021-07-15

**Rating:** 8
**Confidence:** 4

**Summary:**

This paper proposes a DropClass training scheme where class-specific feature maps are used to obtain debased and disentangled representations. This is achieved by randomly dropping out class-specific representation from the feature maps, implemented in an annealing manner, reinforced with a regularization term to further remove class-specific information from the remaining information. Experiments are performed on three segmentation networks on two distinct data sets. The presented method seems to achieves an improvement of performance in most classes, with the most noticeable margin of improvement in minority classes.

**Limitations And Societal Impact:**

The authors adequately addressed the limitations and potential negative societal impact of their work.

**Main Review:**

The presented concept is interesting and original. The methodology is well motivated and clearly described. Experiment results are promising. Some technical questions can be clarified.

First, to make sure that the gradient estimated is reasonable, the authors mentioned that the loss function slowly incorporates DropClass "by linearly increasing the vlaue of lambda from 0 to 1 across the duration of training". How is this linear rate determined exactly, assuming that the training is terminated based on some validation loss and thus the number of epochs needed is not known in advance. Or is the linear ratio pre-determined and irrelevant to the number of training epochs? Furthermore, since it is important to wait until the gradient estimation is reasonably accurate, would some exponential or other types of increase more effective than a linear increase rate?

To avoid heavy computation associated with gradient calculation for each class, the authors focused on convolution layers and implemented the final classification layer using 1x1 convolution as well. Does this mean the presented method will be restricted to convolutional networks?

In experimental results in Pscal, it seems that nearly half to the classes were negatively impacted. What is th implication? Does it suggest that DropClass should be targeting certain classes only rather than all classes, or targeting different classes with different weights? If so, how?

Overall this is a very well written paper that presents an original idea in a simple fashion. It has high discussion potential.

**Time Spent Reviewing:**

2

---

> ### Author Response · Authors · 2021-08-10
> **Response to Reviewer K1Gz**
>
> Thank you so much for the constructive and encouraging comments. Here are our answers for the questions.
>
> **Linear Scaling of $\lambda$**
>
> In our experiments, we train for a fixed number of epochs, so the value of $\lambda$ is a function of the current epoch and the total number of epochs ($\lambda$= curr_epoch / total_epochs). If the training must be terminated based on the validation loss (as mentioned by Reviewer K1Gz), we could consider scaling $\lambda$ linearly until a predetermined epoch, then train with $\lambda = 1$ until the terminating condition is met. We have found that there is no significant difference in the overall performance of either method.
>
> On that note, we believe that the exact choice of function is not so important. As mentioned by R-K1Gz, the main concern is that we allow the gradient estimation to be reasonably accurate before applying the DropClass loss term; then, a wide range of functions --- such as linear or exponential --- could be suitable as long as their outputs are bounded by 0 and 1. For example, when using a 2nd order polynomial function, the overall mIoU increases by 0.1%p compared to the linear scheduling.
>
> ***
> **Is the proposed method restricted to convolutional networks?**
>
> Our proposed method is generic, and not restricted to convolutional networks. We employ 1x1 convolutions as the final classification layer because that is the default setting of the HRNet architecture. Our notes on the gradient calculation were simply meant to be an implementation detail, and this could be easily generalized to other types of layers. We will make this clearer in the final version.
>
> ***
> **Results on Pascal**
>
> In the paper (L223 ~ L241), we hypothesized that the Pascal VOC dataset introduces less bias due to the relatively small number of classes per image (1.5 classes per image). The small number of classes per image implies that there is inherently less bias due to co-location. Furthermore, the class frequency distribution on Pascal is much more uniform than on Cityscapes, which gives us less room for improvement on Pascal.

---

### Official Review · Reviewer_8XQh · 2021-07-16

**Rating:** 6
**Confidence:** 4

**Summary:**

This paper presents DropClass, a new training scheme for reducing feature dependencies among classes to learn debiased and disentangled feature representations. Toward this, the paper
consider Grad-CAM for obtaining the class-specific information, and then randomly discard such information to remove any bias towards a certain class.

A set of experiments on benchmark datasets demonstrate the model’s effectiveness, primarily on under-represented classes.

**Limitations And Societal Impact:**

Yes

**Main Review:**

Strength:
* The proposed training scheme is novel, model-agnostic, and straightforward.
* The experiment covers two crucial datasets in the segmentation literature. Results demonstrate that the inclusion of the proposed training improves over baseline (two well-established frameworks), notably in the under-represented classes.

Weakness:
* The paper aims to remove the issue of feature entanglement and bias vulnerability. These two topics have been well studied in the current ML/DL literature. In my opinion, the presentation of these topics is a bit hand waving. While reading the paper, one gets the impression that these two are interchangeable. The primary goal seems to make sure the model generalizes well across test sets. If so, is this work trying to use disentanglement to achieve debiased representations and then generalization? I think the paper should be clear enough to answer them.
* Lack of comparison against other debiasing techniques. Paper acknowledges and reviews some of them, but no technical comparisons have been made on how such methods would perform in the presented dataset and network architecture. The same applies to disentanglement techniques.
*  The motivation behind L_sup (Eqn 8) is not clear. The impact of Eqn 8 also looks pretty minimal (Table 3).
* The methodological contribution is limited.
* The provided results are not enough to clearly understand the benefit of the proposed training scheme. It is not clear how biased or entangled the baseline framework is before using the proposed training scheme. Only figure 2 is provided to discuss this critical issue.

Questions:
* Authors are encouraged to answer/comment about the points raised in the weakness section.
* Some recent works (Frosst et al. 2019) in disentanglement have pointed out that "maximizing the entanglement of representations of different classes in the hidden layers is beneficial for discrimination in the final layer, possibly because it encourages representations to identify class-independent similarity structures". Can authors discuss them in relation to the proposed training approach?
Frosst, Nicholas, Nicolas Papernot, and Geoffrey Hinton. "Analyzing and improving representations with the soft nearest neighbor loss." International Conference on Machine Learning. PMLR, 2019.
* Isn't this a training procedure? Why does one need to analyze "Test time behavior"?

Grammatical errors:
* Line 140: "encouraged learn debiased" —> "encouraged to learn debiased"

********************* Update **********************
The score has been updated (5 to 6) after the authors' rebuttal.


**Time Spent Reviewing:**

4.5

---

> ### Author Response · Authors · 2021-08-10
> **Response to Reviewer 8XQh**
>
> Thank you for the constructive comments. Your reviews were very helpful to clarify our contribution and improve presentation. Please pay attention to our answers below.
>
> **Clarity regarding disentanglement, debiasing, and generalization**
>
> As mentioned by R-8XQh, the ultimate goal of DropClass is to enhance generalization to the test set. We aim to achieve this by learning disentangled feature representations, which we argue will lead to learning more robust features for semantically similar classes, and alleviate biases among co-located classes (debiasing). Our DropClass method targets both these issues effectively, and we present further experiments (below) to support this argument. We plan to add these experimental results in our final version in hopes that it will clarify our main claim.
> ***
> **Lack of comparison against other debiasing techniques.**
>
> Many works on debiasing focus on the image classification task, and are not easily applicable to the semantic segmentation task. However, as suggested, we provide comparisons with two baseline debiasing techniques: resampling and reweighting.
>
> *Resampling*
>
> Resampling is a bit tricky in semantic segmentation, since the training images often contain multiple classes. For this experiment, we focus solely on the least frequent class of Cityscapes (“motorcycle”). To increase the sampling frequency of the “motorcycle” class, we enlarge the training set such that each image containing the “motorcycle” class is sampled twice at every training epoch. We observe a +7%p increase of IoU from the baseline to 2x motorcycles baseline (motorcycle IoU = 39.9% → 46.9%) for the motorcycle class. But in the case of mIoU, the 2x motorcycles baseline results -0.1%p, meaning that the improvement of the “motorcycle” class comes at the cost of performance on other classes.
> Interestingly, our model still outperforms the 2x motorcycles baseline for both the motorcycle IoU and total mIoU, by +2.8%p, +1.5%p, respectively. The results are shown in Table C below.
>
>
>
> |  | motor. | mIoU|
> |:---:|:---:|:---:|
> | Baseline | 39.9% | 69.2% |
> | Baseline +  2x motor. | 46.9% | 69.1% |
> | DropClass | 49.7% | 70.1% |
> Table C. Resampling (2x motorcycle)
>
> *Reweighting*
>
> In our main paper, both the baseline and DropClass results employ median frequency balancing [1], which is widely used for the Cityscapes dataset. In a nutshell, the cross-entropy term is reweighted according to the frequency of classes.
>
> We conducted some experiments without this reweighting scheme on both the baseline and our DropClass model. The results of this experiment are organized in Table D below:
>
> |  | mIoU | mIoU$^\dagger$ |
> |:---:|:---:|:---:|
> | Baseline w/o reweighting| 69.2% | 57.0% |
> | Baseline | 69.3% | 58.1% |
> | DropClass w/o reweighting | 70.1% | 58.6% |
> | DropClass | 70.7% | 59.7% |
> Table D. Reweighting ablation experiments
>
> As shown in the table, even without the reweighting scheme, the model trained with DropClass outperforms the baseline model with the reweighting scheme.
>
> *References:*
> [1] Badrinarayanan, et al. Segnet: A deep convolutional encoder-decoder architecture for image segmentation. TPAMI, 2017
>
> ***
> **Motivation behind $\mathcal{L}_{\text{sup}}$**
>
> The pixels for the dropped class do not contribute to the loss term, $\mathcal{L}_{\text{CEdrop}}$, and thus, there is no direct supervision for these pixels. Hence, suppressing the probability of the dropped class is required to apply supervision, and it is an essential part of the DropClass method. In Table 3, we observe that the inclusion of this loss term helps the overall mIoU and low-frequency mIoU by 0.5 and 1.1 points, respectively. Given that the improvement from DropClass is 1.4 and 1.6 points respectively, we can see that the suppression term makes up around 50% of the total contribution. Thus, this loss term plays an important role in the overall DropClass method.
>
> ***
> **Benefits of DropClass**
>
> We find that the baseline model is not only prone to learning biases due to the co-location of classes, but also susceptible to learning correlated representations, especially among under-represented classes. Thus, to better illustrate the benefits of DropClass, we have conducted some additional analysis. Please refer to the experimental results presented in our response to Reviewer 4WPr.
>
> We plotted the correlation matrix of the baseline model, where each element corresponds to the cosine similarity between weight vectors of two classes in the final classification layer. We calculated the row-wise non-diagonal sum --- which is a measure of how correlated a class representation is with respect to all other classes --- and observed that more frequently appearing classes tend to be less correlated with other classes. Furthermore, a model trained with DropClass significantly reduces the amount of correlation for all classes. These results are shown in Table A.
>
> Additionally, to examine the effects of co-location, we evaluated the baseline and DropClass models on validation images where the pixels of a given class were “erased”, *i.e.*, replaced with the mean pixel value of the training set. By observing how the IoU of class X is affected by erasing the pixels of class Y, we can quantify the harmful effects of co-location. Moreover, by comparing the performance drops between the baseline and DropClass models, we verify that the DropClass model is less susceptible to the negative effects of co-location during inference. Please refer to Table B for these results.
>
> ***
> **Discussion with Frosst et al. 2019**
>
> Ultimately, both DropClass and the method proposed by Frosst et al. share a similar goal: to enhance the discriminative power of the model. Frosst et al. achieve this by encouraging entanglement in the hidden layers, while we employ a class drop method. With that said, we are skeptical whether encouraging feature entanglement in the early layers can benefit the semantic segmentation task. Since segmentation datasets contain multiple classes per image, we suspect that explicitly entangling features may have an adverse effect on the model’s ability to identify class boundaries (within a given image). We will add this discussion in the related works section of our final paper.
>
> ***
> **Test time behavior**
>
> Yes, the paragraph title is a misnomer. DropClass is a training procedure, and we do not need to analyze the “test time behavior”. This paragraph was meant to emphasize that our method does not require the true gradient (using ground truth labels, $y^c$). We apologize for the confusion and will update the paper accordingly.

---

> > ### Comment · Reviewer_8XQh · 2021-08-25
> > **Response to authors' rebuttal.**
> >
> > I thank the authors for responding to my comments. The added experiments, especially the comparison with debasing techniques, are helpful to clear out some of the concerns that I had.
> >
> > In light of these new experiments and hoping, authors would change the paper as discussed, I will be updating my score to 6, marginally above the acceptance threshold.

---

> > > ### Author Response · Authors · 2021-08-31
> > > **Re: Response to authors' rebuttal.**
> > >
> > > Dear Reviewer 8XQh,
> > >
> > > Thank you so much for your encouraging comments. We will revise our paper as discussed and improve the presentation.
> > >
> > > Best wishes,
> > >
> > > Authors

---

### Official Review · Reviewer_4WPr · 2021-07-18

**Rating:** 6
**Confidence:** 5

**Summary:**

Recent semantic segmentation methods often learn biased representation from the dataset with highly co-located objects (e.g., motorcycle and road) as well as the class imbalance. This paper proposes a training scheme of dropping the class-specific representations in the highly-entangled features, mitigating the bias induced by the dropped class. The proposed DropClass is also model-agnostic and demonstrates the improved segmentation results in Cityscapes and Pascal VOC datasets.

**Limitations And Societal Impact:**

Yes. The authors included the limitations and negative societal impact in Section 5.

**Main Review:**

Strength
1)	The paper is well organized and easy to follow.
2)	The main problem setting of the dataset bias in the semantic segmentation task is novel.
3)	The improved IoU scores, particularly for the under-represented classes, are significant.

Weakness
I think the overall experimental settings and results only show the improved performance on under-represented classes without evaluating the debiased performance between highly correlated classes, which is the main topic addressed in this paper.
-	Tables 1 and 2 do not contain the information on which classes suffer from the bias. I am concerned that the low class frequency does not necessarily cause the classes to be co-located with other classes. In other words, frequently appearing classes may also suffer from the performance degradation due to the highly correlated object classes.
-	Similarly, are Train-Truck, Truck-Bus, and Bicycle-Motorcycle highly co-located objects in the Cityscapes dataset? Does Vanilla fail because they have the correlation between the classes or because the frequency of such classes is low?

I would like the authors to 1) specify the classes vulnerable to the bias due to the unwanted correlation with other classes and 2) provide the quantitative comparison between classes with bias and without bias. (e.g., class A is often highly co-located with class B, and the segmentation results of A are xxx worse compared to the class C which does not have the unwanted correlation with other classes.) Additionally, the authors need to demonstrate the improved results of these classes by the proposed DropClass.

**Time Spent Reviewing:**

48 hours

---

> ### Author Response · Authors · 2021-08-10
> **Response to Reviewer 4WPr**
>
> Thank you for the constructive comments, especially about the suggestions for some experiments. Please read our feedback and let us know if you have any further questions.
>
> **Does bias vulnerability come from low class frequency?**
>
> It is true that low class frequency does not necessarily translate to co-location with other classes; for example, the “truck” and “train” classes only appear together in 7 of the 2975 training images. However, we do observe that classes with low frequency are clearly more correlated with other classes in the baseline model. This observation was made by plotting the $N_{\text{class}} \times N_{\text{class}}$ correlation matrix of class weights in the final 1x1 convolutional layer of the baseline model, where the (i, j)-th element corresponds to the cosine similarity of the i-th and j-th class vectors (the full plot will be included in the supplementary materials). We notice that the most frequent “road” class has low similarities with all classes, while the least frequent “motorcycle” class has high similarities with multiple classes, such as “bicycle” and “rider”. To this end, the row-wise non-diagonal sum of correlations among classes is presented in Table A below. Note that there is a clear trend between increasing frequency and decreasing correlations among classes.
>
> |  | motor. | rider | t.light | train | bus | truck | bicyc. | t.sign | wall | fence |
> |:---:|:---:|:---:|:---:|:---:|:---:|:---:|:---:|:---:|:---:|:---:|
> | Freq (%) | 0.1 | 0.1 | 0.2 | 0.2 | 0.2 | 0.3 | 0.4 | 0.6 | 0.7 | 0.9 |
> | Corr. | 1.24 | 1.20 | 0.97 | 0.65 | 0.91 | 0.88 | 0.85 | 0.60 | 0.46 | 0.45 |
> | $\Delta$  | 0.50 | 0.43 | 0.32 | 0.18 | 0.28 | 0.30 | 0.33 | 0.21 | 0.22 | 0.22 |
>
> |  | fence | terrain | person | pole | sky | s.walk | car | veg. | building | road |
> |:---:|:---:|:---:|:---:|:---:|:---:|:---:|:---:|:---:|:---:|:---:|
> | Freq (%) | 0.9 | 1.2 | 1.2 | 1.2 | 4.0 | 6.1 | 7.0 | 15.9 | 22.8 | 36.9 |
> | Corr. | 0.45 | 0.53 | 0.48 | 0.63 | 0.16 | 0.45 | 0.30 | 0.35 | 0.33 | 0.20 |
> | $\Delta$  | 0.22 | 0.27 | 0.17 | 0.35 | 0.08 | 0.20 | 0.11 | 0.19 | 0.24 | 0.08 |
>
> Table A. Class correlations
>
> Furthermore, we observe that the model trained with DropClass shows lower inter-class correlations for all class pairs, as summarized in the last row of Table A ($\Delta$ indicates correlations of the baseline model subtracted by correlations of the DropClass model).
>
> Our proposed training scheme aims to decrease the inter-class dependencies, which leads to learning strong class-specific representations. This is reflected in our correlation analysis above (Table A), and also in the improved performance, especially for under-represented classes.
>
> ***
> **Specify the classes vulnerable to the bias with quantitative comparison**
>
> It is difficult to rigorously define and analyze the bias vulnerability of the classes in real-world segmentation datasets. Instead, we conduct a simple experiment to observe effects of co-location and correlation between classes.
>
> The baseline and DropClass models are evaluated on the validation images (I*), where the pixel values of class X are “erased”, i.e., replaced with mean RGB values of the training set. By observing how the IoU of class Y is affected by “erasing” the pixels of class X, we can quantify the harmful effects of co-location ($\Delta$). Furthermore, we can determine which model is less susceptible to the negative effects of co-location by comparing the $\Delta$ values of the two models. A few representative examples are shown in Table B, where the middle three columns pertain to the baseline model, and the rightmost three columns pertain to our DropClass model.
>
> | X | Y | I | I* | $\Delta$ | I | I* | $\Delta$ |
> |:---:|:---:|:---:|:---:|:---:|:---:|:---:|:---:|
> | road | person | 81.8% | 69.6% | -12.2% | 82.6% | 74.5% | -8.1% |
> | road | car | 94.5% | 88.0% | -6.5% | 94.9% | 90.6% | -4.4% |
> | car | person | 81.0% | 57.0% | -23.9% | 81.9% | 69.8% | -12.1% |
> | truck | train | 75.2% | 71.6% | -3.6% | 77.8% | 77.8% | 0.0% |
>
> Table B. Quantifying the effects of co-location. X is the removed class, Y is the evaluated class
>
> As shown in the “road-person”, “road-car”, “car-person” pairs, the baseline model suffers from biases induced by the spatial relations of the class pairs. In contrast, the DropClass model alleviates these biases, as indicated by the smaller drop in IoU.
>
> While the “truck-train” classes are not necessarily co-located, they are semantically similar classes. By removing all pixels of the “truck” class, we can observe how the features of “truck” affect those of the “train” class. For the baseline model, we notice that the “train” IoU decreases by 3.6%p, which suggests that the features of “train” are entangled with the features of “truck”, to some extent. On the other hand, the “train” features of the DropClass model seems unaffected by the absence of “truck”, which suggests that DropClass has successfully disentangled feature representations.

---

> > ### Comment · Reviewer_4WPr · 2021-08-20
> > **Response to Authors' Rebuttal**
> >
> > I would like to thank the authors for responding to my concerns with additional experiments. However, I am not still convinced that the paper properly tackles the bias problem for the following reasons.
> >
> > 1. High correlation of class weights of two different classes does not necessarily indicate the strong co-location of such classes in the training dataset. Rather, such high correlation can be due to the low frequency of the classes themselves. In other words, the class of objects with a low frequency, e.g., motorcycle, may lead the classifier to misclassifying them into another similar class  with a high frequency, e.g., bicycle. This causes the class weights of such two classes to be less discriminative, resulting in the high cosine similarity between them.
> >
> > 2. As the author mentioned that the truck-train classes are not necessarily co-located but are semantically similar classes, the improved results of “truck-train” is not caused by resolving the bias or the strong co-location of truck and train. Rather, as they are semantically similar classes, the features for the minor class can be easily entangled with those of the other major class. Therefore, such a result demonstrates that the DropClass achieves the disentangled representation between truck and train, but not the de-biased one for these classes.
> >
> > 3. The submitted paper does not include the evaluation of the model on the “unbiased” test set. Comparison of the IoU scores of both baselines and DropClass in the unbiased test set should be included. I think the experiments of erasing the co-located objects presented in the authors’ response can be one of the novel approaches of creating the unbiased test set. In this regard, I agree that the results of “road-person”, “road-car”, and “car-person” demonstrate that DropClass can alleviate the biased representation for the highly co-located objects to a certain degree, resulting in improved results in the unbiased test set.
> >
> > In conclusion, I think the paper should clarify the motivation and thorough definition of the bias in the semantic segmentation task along with additional supporting evidence, such as quantitative analysis on the co-location of different classes and corresponding performance decrease in the unbiased test set.

---

> > > ### Author Response · Authors · 2021-08-25
> > > **Response to Reviewer 4WPr**
> > >
> > > We are thankful for your detailed response and feedback again.
> > >
> > > The representations of a class are often entangled with others due to the semantic similarity between the classes and/or the frequent co-location (or co-occurrence) of two or more classes. In both cases, rare classes typically become the victims of the feature entanglement. In this sense, "dataset bias" and "feature entanglement" are not mutually exclusive, but dataset bias is manifested as feature entanglement. Thus, we argue that the issues caused by dataset bias can be alleviated via feature disentanglement. DropClass aims to reduce dataset biases by breaking inter-class relationships in the highly entangled feature representation space.
> > >
> > > To better highlight the debiasing effects of DropClass, we employ an “unbiased” test set (I*), where each of the 19 classes in Cityscapes is erased from the original test set (I). Since it is difficult to rigorously quantify the co-location of classes, we define a simple proxy instead: the drop in IoU for each class pair (e.g., IoU of “person” after erasing “road”) of the baseline model. For example, the “person” class is most affected by the absence of “road”, “sidewalk” and “bicycle” classes, which suggests that these classes are highly co-located with the “person” class. For each evaluated class, we average IoUs after removing the Top3 most influential classes. The same sets of Top3 classes are used to evaluate the DropClass model on the “unbiased” test set, and we present the results in Table E below.
> > >
> > > |               |  rider | motor. |  train |  wall  | t.light | bicyc. |  bus  | person |  pole | s.walk |
> > > |:-------------:|:------:|:------:|:------:|:------:|:-------:|:------:|:-----:|:------:|:-----:|:------:|
> > > |  Baseline(I)  |  61.7% |  61.9% |  72.6% |  59.1% |  72.0%  |  76.9% | 89.2% |  82.5% | 67.8% |  85.5% |
> > > |  Baseline(I*) |  33.0% |  37.8% |  53.9% |  40.5% |  59.2%  |  66.5% | 80.3% |  74.3% | 60.6% |  79.3% |
> > > |    $\Delta$   | -28.7% | -24.1% | -18.7% | -18.6% |  -12.8% | -10.4% | -9.0% |  -8.3% | -7.2% |  -6.2% |
> > > |  DropClass(I) |  61.7% |  61.9% |  72.6% |  59.1% |  72.0%  |  76.9% | 89.2% |  82.5% | 67.8% |  85.5% |
> > > | DropClass(I*) |  34.0% |  43.8% |  61.0% |  45.8% |  62.1%  |  67.1% | 82.6% |  76.3% | 65.2% |  83.0% |
> > > |    $\Delta$   | -27.7% | -18.1% | -11.6% | -13.3% |  -9.8%  |  -9.8% | -6.6% |  -6.2% | -2.7% |  -2.5% |
> > >
> > > |               | fence | terrain | truck |  sky  |  car  |  veg. | building | t.sign |  road |  mIoU |  mIoU* |
> > > |:-------------:|:-----:|:-------:|:-----:|:-----:|:-----:|:-----:|:--------:|:------:|:-----:|:-----:|:------:|
> > > |  Baseline(I)  | 63.1% |  64.8%  | 72.9% | 95.2% | 94.8% | 92.0% |   92.7%  |  79.1% | 98.1% | 78.0% |  71.5% |
> > > |  Baseline(I*) | 57.7% |  60.1%  | 68.6% | 91.8% | 92.2% | 89.4% |   90.0%  |  76.7% | 97.9% | 68.9% |  56.2% |
> > > |    $\Delta$   | -5.4% |  -4.7%  | -4.2% | -3.4% | -2.7% | -2.6% |   -2.6%  |  -2.4% | -0.2% | -9.1% | -15.3% |
> > > |  DropClass(I) | 63.1% |  64.8%  | 72.9% | 95.2% | 94.8% | 92.0% |   92.7%  |  79.1% | 98.1% | 78.0% |  71.5% |
> > > | DropClass(I*) | 57.1% |  59.1%  | 66.8% | 92.4% | 93.5% | 90.3% |   90.4%  |  77.9% | 97.9% | 70.9% |  59.8% |
> > > |    $\Delta$   | -6.0% |  -5.7%  | -6.0% | -2.7% | -1.4% | -1.6% |   -2.3%  |  -1.2% | -0.2% | -7.1% | -11.7% |
> > >
> > > Table E. Quantifying the effects of bias. Top3 most influential classes are selected from the baseline model. mIoU* indicates mIoU for the 9 classes with the highest bias vulnerability.
> > >
> > >
> > > As seen in Table E, the baseline model obtains an mIoU of 68.9, while our DropClass model obtains an mIoU of 70.9. Across the majority classes, we observe that the DropClass model suffers from less performance degradation. Also, note that the “rider” class, by definition, is dependent on the presence of “motorcycle” or “bicycle”, which may rationalize the reason behind the smaller difference of $\Delta$ between the baseline and DropClass models.
> > >
> > > Finally, in Table F below we present results for the same experiment, except that the Top3 class selection scheme is based on the DropClass model.
> > >
> > > |               |  rider | motor. |  wall  |  train | bicyc. | t.light | truck |  bus  | fence | person |
> > > |:-------------:|:------:|:------:|:------:|:------:|:------:|:-------:|:-----:|:-----:|:-----:|:------:|
> > > |  Baseline(I)  |  62.6% |  60.5% |  54.9% |  67.8% |  76.5% |  71.0%  | 73.3% | 87.5% | 62.6% |  81.7% |
> > > |  Baseline(I*) |  33.0% |  37.8% |  40.5% |  54.2% |  66.5% |  59.2%  | 69.0% | 80.3% | 58.0% |  74.3% |
> > > |    $\Delta$   | -29.6% | -22.7% | -14.4% | -13.5% | -10.0% |  -11.8% | -4.3% | -7.3% | -4.6% |  -7.5% |
> > > |  DropClass(I) |  61.7% |  61.9% |  59.1% |  72.6% |  76.9% |  72.0%  | 72.7% | 89.2% | 63.2% |  82.5% |
> > > | DropClass(I*) |  34.0% |  43.8% |  45.8% |  60.0% |  67.1% |  62.1%  | 65.3% | 82.6% | 56.8% |  76.3% |
> > > |    $\Delta$   | -27.7% | -18.1% | -13.3% | -12.6% |  -9.8% |  -9.8%  | -7.4% | -6.6% | -6.4% |  -6.2% |
> > >
> > > |               | terrain |  sky  |  pole | s.walk | building |  veg  |  car  | t.sign |  road |  mIoU |  mIoU* |
> > > |:-------------:|:-------:|:-----:|:-----:|:------:|:--------:|:-----:|:-----:|:------:|:-----:|:-----:|:------:|
> > > |  Baseline(I)  |  66.2%  | 94.3% | 63.3% |  85.9% |   92.2%  | 92.4% | 94.4% |  77.7% | 98.2% | 77.0% |  68.5% |
> > > |  Baseline(I*) |  60.1%  | 91.8% | 60.6% |  79.4% |   90.0%  | 90.3% | 92.2% |  76.7% | 97.9% | 69.0% |  55.4% |
> > > |    $\Delta$   |  -6.2%  | -2.6% | -2.7% |  -6.6% |   -2.1%  | -2.2% | -2.2% |  -1.1% | -0.3% | -8.0% | -13.1% |
> > > |  DropClass(I) |  64.8%  | 95.2% | 67.8% |  85.6% |   92.7%  | 92.7% | 94.8% |  79.1% | 98.1% | 78.0% |  69.9% |
> > > | DropClass(I*) |  59.1%  | 92.4% | 65.2% |  83.0% |   90.4%  | 91.0% | 93.5% |  77.9% | 97.9% | 70.8% |  57.5% |
> > > |    $\Delta$   |  -5.7%  | -2.7% | -2.7% |  -2.6% |   -2.3%  | -1.8% | -1.4% |  -1.2% | -0.2% | -7.3% | -12.4% |
> > >
> > > Table F. Quantifying the effects of bias. Top3 most influential classes are selected from the DropClass model. mIoU* indicates mIoU for the 9 classes with the highest bias vulnerability.
> > >
> > >
> > > We hope these results have cleared up any confusion and uncertainty regarding our arguments in the main paper. Again, we thank Reviewer 4WPr for the feedback, and will add these results into the main paper to demonstrate the debiasing effects of DropClass.

---

> > > > ### Comment · Reviewer_4WPr · 2021-08-26
> > > > **Response to Authors' Rebuttal**
> > > >
> > > > 1. I would like to thank the authors for demonstrating the debiasing capability of DropClass on the “unbiased” test set in the experiment setup I suggested. The results are helpful for understanding the effectiveness of DropClass on alleviating the biased representation due to the co-location of different objects in the scene.
> > > >
> > > > 2. I partially agree with the authors’ statement that ‘dataset bias’ and ‘feature entanglement’ are not mutually exclusive. However, as far as I know, the scope of the dataset bias problem addressed in the previous literature on image classification or VQA task [1,2,3,4,5] does not cover the semantic similarity. For example, when the biased classifier classifies the ‘ship’ by looking at the blue background, it will classify the airplane with the blue sky as ‘ship’. Such failure is due to the correlation between ship and background, not the semantic similarity between ship and airplane. Such entangled representation due to the semantic similarity is more related to the long-tailed classification where objects of minor classes are often misclassified into the similar ones in the major classes. Therefore, I would like to ask the authors to cite the previous studies assuming such semantic similarity as the dataset bias, and the authors need to clarify the definition of the dataset bias addressed in their paper.
> > > >
> > > > [1] Geirhos et al., “Shortcut Learning in Deep Neural Networks”, Nature Machine Intelligence
> > > >
> > > > [2] Wang et al., “Learning Robust Representations by Projecting Superficial Statistics Out”, ICLR’19
> > > >
> > > > [3] Cadene et al., “RUBi: Reducing Unimodal Biases for Visual Question Answering”, NeurIPS’19
> > > >
> > > > [4] Bahng et al., “Learning De-biased representations with Biased representations”, ICML’20
> > > >
> > > > [5] Nam et al., “Learning from Failure: Training Debiased Classifier from Biased Classifier”, NeurIPS’20

---

> > > > > ### Author Response · Authors · 2021-08-28
> > > > > **Response to Reviewer 4WPr**
> > > > >
> > > > >  First of all, we thank you for your detailed comments about the issues related to bias and understand your point. However, dataset biases in the segmentation task are rather distinct from the image classification and VQA task. While a “blue background” would serve as a bias attribute in classification (affecting “ship” and “airplane”), in segmentation, it would serve as a different class. Thus, biases can be caused by inter-class relationships in the segmentation task. Consequently, we also consider the fact that mixed (correlated) features of different classes could result in unwanted biases among classes.
> > > > >  We hope that the experiment in our response to 8XQh, which compares DropClass with other debasing techniques (resampling, reweighting), will help in further understanding the characteristics of DropClass for debiasing. We will update the main paper to clarify the definition of the dataset bias addressed in DropClass with discussions of the papers you mentioned. Again, we truly appreciate the time you have taken to evaluate the study, as well as your timely review.

---

> > > > > > ### Comment · Reviewer_4WPr · 2021-08-30
> > > > > > **Response to Authors' Rebuttal**
> > > > > >
> > > > > > I would like to thank the authors for the responses addressing my concern. As the dataset bias problem in the semantic segmentation task has been under-explored, I believe that the thorough clarification on the definition as well as the characteristic of the dataset bias is significantly important in such a task, which is insufficient in this paper. Yet, the additional results including the evaluation on ‘unbiased test set’ have demonstrated the debiasing capability of DropClass and I would like to update my score to 6, marginally above the acceptance threshold. Additionally, I strongly recommend the authors to include 1) the definition of the dataset bias, 2) the discussion on the relationship between the co-location of different classes and their entangled representations, and 3) additional comparisons to the baselines tackling the entangled representations between semantically similar classes.

---

> > > > > > > ### Author Response · Authors · 2021-08-31
> > > > > > > **Re: Response to Authors' Rebuttal**
> > > > > > >
> > > > > > > Dear Reviewer 4WPr,
> > > > > > >
> > > > > > > Your comments are indeed helpful to improve our paper. We will reflect on your feedback, especially for formalizing the dataset bias issues in semantic segmentation and clarifying the benefits of DropClass via thorough experiments.
> > > > > > >
> > > > > > > Thank you so much!
> > > > > > >
> > > > > > > Authors

---

### Author Response · Authors · 2021-08-10
**To All Reviewers**

We would like to thank all reviewers for their insightful comments. For each review, we answer individual questions and present results for additional analysis that will help strengthen our submission.

---

### Decision · Program_Chairs · 2021-09-27

**Decision:**

Accept (Poster)

**Comment:**

This work describes a new training scheme for semantic segmentation models (“DropClass”) that is intended to encourage the learning of feature representations that avoid entangling features from objects (e.g. car & road) that tend to co-occur/co-locate.

Reviewers were generally positive about the work, and found it to be interesting and novel. Reviewers and authors had something of a philosophical debate about the nature of dataset bias and entanglement, and it will be helpful for the authors to clarify these terms in the resulting paper, and to ensure that limitations of the method (e.g. tendency for low frequency classes to become entangled with semantically similar class) are appropriately highlighted for readers. All in all, this is strong work that should be of interest to the NeurIPS community.